# Neural Implicit Shape Editing using Boundary Sensitivity

**Arturs Berzins**
Department of Mathematics and Cybernetics
SINTEF
`arturs.berzins@sintef.no`

**Moritz Ibing & Leif Kobbelt**
Visual Computing Institute
RWTH Aachen University

## Abstract

Neural fields are receiving increased attention as a geometric representation due to their ability to compactly store detailed and smooth shapes and easily undergo topological changes. Compared to classic geometry representations, however, neural representations do not allow the user to exert intuitive control over the shape. Motivated by this, we leverage *boundary sensitivity* to express how perturbations in parameters move the shape boundary. This allows to interpret the effect of each learnable parameter and study achievable deformations. With this, we perform *geometric editing*: finding a parameter update that best approximates a globally prescribed deformation. Prescribing the deformation only locally allows the rest of the shape to change according to some prior, such as *semantics* or *deformation* rigidity. Our method is agnostic to the model its training and updates the NN in-place. Furthermore, we show how boundary sensitivity helps to optimize and constrain objectives (such as surface area and volume), which are difficult to compute without first converting to another representation, such as a mesh.

## 1 Introduction

A *neural field* is a neural network (NN) mapping every point in a domain of interest, typically of 2 or 3 dimensions, to one or more outputs, such as a signed distance function (SDF), occupancy probability, opacity or color. This allows to represent smooth, detailed, and watertight shapes with topological flexibility, while being compact to store compared to classic implicit representations (Davies et al., 2020). When the NN is trained not on a single shape but instead an entire collection, each shape is encoded in a latent vector, which is an additional input to the NN (Park et al., 2019; Chen & Zhang, 2019; Mescheder et al., 2019). As a result, neural fields are receiving increased interest as a geometric representation in numerous applications, such as shape generation (Park et al., 2019), shape completion (Chibane et al., 2020), shape optimization (Remelli et al., 2020), scene representation (Sitzmann et al., 2020), and view synthesis (Mildenhall et al., 2020). Some pioneering works have also investigated geometry processing, like smoothing and deformation, on neural implicit shapes (Yang et al., 2021; Remelli et al., 2020; Mehta et al., 2022; Guillard et al., 2021), but these can be computationally costly or resort to intermediate mesh representations. In part, this difficulty stems from the shape being available only *implicitly* as the sub-level set of the field.

While intuitive (often synonymous with local) geometric control is a key design principle of classic explicit or parametric representations (like meshes, splines, or subdivision schemes), it is not trivial to edit even classic implicit representations, especially ones with global functions (Bærentzen & Christensen, 2002). Previous works on neural implicit shape editing have focused on the shape *semantics*, i.e. changing part-level features based on the whole shape structure, but achieve this through tailored training procedures or architectures or resort to intermediate mesh representations.

We instead propose a framework which unifies geometric and semantic editing and which is agnostic to the model and its training and modifies the given model in-place akin to classic representations. To treat the geometry, not the field, as the primary object we consider *boundary sensitivity* to relate changes in the function parameters and the implicit shape. This allows us to express and interpret a basis for the displacement space.

In this framework, the user supplies a target displacement on (a part) of the shape boundary in the form of deformation vectors or normal displacements at a set of surface points. Employing boundary sensitivity we find the parameter update which best approximates the prescribed deformation. In *geometric* editing, we prescribe an exact geometric update on the entirety of the boundary. Akin to local control in classic representations, we especially study the case where the prescribed displacement is local and the rest of the boundary is fixed. In *semantic* editing only a part of the boundary is prescribed a target displacement. The remaining unconstrained displacement is determined by leveraging the generalization capability of the model as an additional prior, producing semantically consistent results on the totality of the shape. Another prior often used in shape editing is based on *deformation* energy, such as as-rigid-as-possible (Sorkine & Alexa, 2007) or as-Killing-as-possible (Solomon et al., 2011), which generates physically plausible deformations minimizing stretch and bending. This is not trivially applicable to implicit surfaces, as there is no natural notion of stretch due to ambiguity in tangent directions. We discuss a few options to resolve this ambiguity and demonstrate that boundary sensitivity can be leveraged to optimize directly in the space of expressible deformations. Lastly, we use level-set theory and boundary sensitivity to constrain a class of functionals, which are difficult to compute without first converting to another representation, such as a mesh. As a specific use-case, we consider fixing the volume of a shape to prevent shrinkage during smoothing.

## 2  RELATED WORK

**Implicit Shape Representations and Manipulation**  Implicit shape representations or *level-sets* have been widely used in fields such as computer simulation (Sethian & Smereka, 2003), shape optimization (van Dijk et al., 2013), imaging, vision, and graphics (Vese, 2003; Tsai & Osher, 2003). Classically, an implicit function is represented as a linear combination of either a few global basis functions, such as in *blobby models* (Muraki, 1991), or many local basis functions supported on a (potentially adaptive) volumetric grid (Whitaker, 2002). While few global bases use less memory, the task of expressing local displacements is generally ill-posed (Whitaker, 2002). Hence, methods for interactive editing of implicit shapes are formulated for grid-supported level-sets (Museth et al., 2002; Bærentzen & Christensen, 2002). These methods use a prescribed velocity field, for which the *level-set-equation* – a partial differential equation (PDE) modelling the evolution the surface – is solved using numerical schemes on the discrete spatiotemporal grid.

**Neural Fields**  In *neural fields*, a NN is used to represent an implicit function. Different from classic implicit representations, these are non-linear and circumvent the memory-expressivity dilemma Davies et al. (2020). In addition, automatic differentiation also provides easy access to differential surface properties, such as normals and curvatures, useful for smoothing and deformation (Yang et al., 2021; Mehta et al., 2022; Atzmon et al., 2021) or more exact shape fitting (Novello et al., 2022). Early works propose to use vanilla multilayer perceptrons (MLPs) to learn occupancy probability (Mescheder et al., 2019; Chen & Zhang, 2019) or signed distance functions (Park et al., 2019; Atzmon & Lipman, 2020), whose level-sets define the shape boundary. Conditioning the NN on a latent code as an additional input allows to decode a collection of shapes with a single NN (Chen & Zhang, 2019; Mescheder et al., 2019; Park et al., 2019). Later works build upon these constructions by introducing a spatial structure on the latent codes using (potentially adaptive) grids, which affords more spatial control when generating novel shapes (Ibing et al., 2021) and allows to reconstruct more complex shapes and scenes (Jiang et al., 2020; Peng et al., 2020; Chibane et al., 2020). In this work, we develop a method to interactively modify shapes generated by any of these methods.

**Neural Shape Manipulation**  Although there are many previous works on the deformation of shapes with NNs, we focus only on methods that use neural implicit representations. These can roughly be sorted into two groups based on their guiding principle. Methods in the first group manipulate shapes based on a *semantic* principle. Hertz et al. (2022) create a generative framework with part-level control consisting of three NNs decomposing and mixing shapes in the latent space. Elsner et al. (2021) encourage the latent code to act as geometric control points, allowing to manipulate the geometry by moving the control points. Chen et al. (2021) demonstrate how to interpolate between shapes while balancing their semantics by choosing which layer's features to track. Similar to our method, this is agnostic to the training and the model. Hao et al. (2020) learn a joint latent space between an SDFs and its coarse approximation in the form of a union of spherical primitives. This way modifications of the spheres can be translated to the best matching change of the high-fidelity shape.

In our work, we allow a similar editing approach, but through insights from boundary sensitivity, we are able to apply such manipulation to arbitrary NNs.

Methods in the second group are based on some well defined *energy*. Yang et al. (2021) demonstrate shape smoothing using a curvature-based loss. Similarly, shape deformation is performed by optimizing the deformation energy for which another invertible correspondence NN is needed. Niemeyer et al. (2019) train an additional NN as a spatio-temporally continuous velocity field, along which the points of a shape are integrated.

**Boundary Sensitivity**   Boundary sensitivity has already been introduced in the context of implicit neural shapes in several previous works. Atzmon et al. (2019) use it to translate geometric losses defined on points sampled from a classifier or SDF level-sets to parameter updates during training. Atzmon et al. (2021) use approximate Killing vector fields (AKVFs) and boundary sensitivity during training to encourage latent space interpolation to act as physically plausible deformation. Neural implicit shape manipulation has also been achieved by leveraging meshes as an intermediate representation to benefit from the well studied geometry processing operations on those. Remelli et al. (2020) propose differentiable mesh extraction to then propagate gradients from a differentiable operation on the mesh through to the implicit NN. Mehta et al. (2022) study this link in more detail using classic theory of level-sets, using boundary sensitivity to translate several mesh-based operations to SDF updates. Guillard et al. (2021) builds upon differentiable mesh extraction for sketch-based editing by translating an image-based error to shape updates. Similar to our work, Sketch2Mesh uses boundary sensitivity only for the editing, being model and training agnostic. While our method trivially generalizes to meshes, we require only points sampled from the surface. We further show how to restrict the editing process by introducing explicit constraints or by using the NN's semantic prior.

## 3   BOUNDARY SENSITIVITY

Let $f$ be a differentiable function mapping a spatial coordinate $\mathbf{x} \in \mathcal{D} \subset \mathbb{R}^D$ to a scalar $f(\mathbf{x}; \boldsymbol{\Theta}) \in \mathbb{R}$ on a domain of interest $\mathcal{D}$. Let the vector $\boldsymbol{\Theta} \in \mathbb{R}^P$ gather the parameters of $f$. For a given $\boldsymbol{\Theta}$, the sign of $f$ *implicitly* defines the shape $\Omega(\boldsymbol{\Theta}) = \{\mathbf{x} \in \mathcal{D} | f(\mathbf{x}; \boldsymbol{\Theta}) \leq 0\}$ and its boundary $\Gamma(\boldsymbol{\Theta}) = \{\mathbf{x} \in \mathcal{D} | f(\mathbf{x}; \boldsymbol{\Theta}) = 0\}$.

Consider the variation of $\Omega(\boldsymbol{\Theta})$ by a displacement field (also known as *velocity field*) $\delta\mathbf{x} : \mathcal{D} \mapsto \mathbb{R}^D$, denoted by $\Omega_{\delta\mathbf{x}}(\boldsymbol{\Theta}) = \{\mathbf{x} + \delta\mathbf{x}(\mathbf{x}) \in \mathcal{D} | f(\mathbf{x} + \delta\mathbf{x}(\mathbf{x}); \boldsymbol{\Theta}) \leq 0\}$. If $\delta\mathbf{x}$ is sufficiently small, the initial and the perturbed shapes are diffeomorphic (Allaire et al., 2004), meaning there is a one-to-one correspondence between the points of both. Consider the displacement field $\delta\mathbf{x}$ induced by a sufficiently small parameter perturbation $\delta\boldsymbol{\Theta}$ and the perturbed shape $\Omega_{\delta\mathbf{x}}(\boldsymbol{\Theta}) = \Omega(\boldsymbol{\Theta} + \delta\boldsymbol{\Theta})$. To compute $\delta\mathbf{x}$, consider the total derivative of the boundary condition $f(\mathbf{x}; \boldsymbol{\Theta}) = 0$:

$$\mathrm{d}f(\mathbf{x}; \boldsymbol{\Theta}) = \nabla_{\mathbf{x}}f^\top \, \mathrm{d}\mathbf{x} + \nabla_{\boldsymbol{\Theta}}f^\top \, \mathrm{d}\boldsymbol{\Theta} = 0 \quad \forall \mathbf{x} \in \Gamma \, . \tag{1}$$

We can replace the infinitesimal increments $\mathrm{d}\mathbf{x}$ and $\mathrm{d}\boldsymbol{\Theta}$ with sufficiently small perturbations $\nabla_{\mathbf{x}}f^\top \delta\mathbf{x} + \nabla_{\boldsymbol{\Theta}}f^\top \delta\boldsymbol{\Theta} = 0$. Assuming bounded gradients $\nabla_{\mathbf{x}}f = \nabla_{\mathbf{x}}f(\mathbf{x}; \boldsymbol{\Theta}) \in \mathbb{R}^D$ and $\nabla_{\boldsymbol{\Theta}}f = \nabla_{\boldsymbol{\Theta}}f(\mathbf{x}; \boldsymbol{\Theta}) \in \mathbb{R}^P$ allows to express $\delta\mathbf{x} = \delta\mathbf{x}(\mathbf{x}; \boldsymbol{\Theta}, \delta\boldsymbol{\Theta})$ as

$$\delta\mathbf{x} = \frac{-\nabla_{\mathbf{x}}f \nabla_{\boldsymbol{\Theta}}f^\top \delta\boldsymbol{\Theta}}{\|\nabla_{\mathbf{x}}f\|^2} + \delta\mathbf{x}_t \, . \tag{2}$$

This is what we refer to as *boundary sensitivity* – the movement of the boundary $\delta\mathbf{x}$ caused by the parameter perturbations $\delta\boldsymbol{\Theta}$. Equation 2 consists of normal and tangent components $\delta\mathbf{x} = \delta\mathbf{x}_n + \delta\mathbf{x}_t = \mathbf{n}\delta x_n + \mathbf{t}\delta x_t$. With $\mathbf{n} = \nabla_{\mathbf{x}}f/\|\nabla_{\mathbf{x}}f\|$ the outward facing normal, we rewrite the normal part as the weighted sum of the *basis* $\mathbf{b} = \mathbf{b}(\mathbf{x}; \boldsymbol{\Theta}) \in \mathbb{R}^P$

$$\delta x_n = \mathbf{b}^\top \delta\boldsymbol{\Theta} \quad , \quad \mathbf{b} = \frac{-\nabla_{\boldsymbol{\Theta}}f}{\|\nabla_{\mathbf{x}}f\|} \, . \tag{3}$$

For each positive parameter perturbation $\delta\Theta_p, p \in \{1..P\}$, a positive gradient $\partial f/\partial \Theta_p$ implies that the boundary moves inward along the normal, since the value of $f$ at $\mathbf{x}$ increases. The total movement

caused by all parameter perturbations is their superposition. We show some basis functions in Figures 2 and 3.

The tangential component $\delta\mathbf{x}_t$ is ambiguous since per definition $\nabla_{\mathbf{x}}f^\top\delta\mathbf{x}_t = 0$. This is transport of the surface along itself which has no effect on the implicit representation. However, this ambiguity can be resolved to recover $\delta\mathbf{x}_t$ based on an additional assumption. In the context of classic implicit representations two such assumptions considered are that normals of points do not change during the deformation (Jos & Schmidt, 2011) and that they are nearly-isometric (Tao et al., 2016).

**Editing** Let $\delta\bar{\mathbf{x}} : \bar{\Gamma} \mapsto \mathbb{R}^D$ be a prescribed displacement on (a part of) the boundary $\bar{\Gamma} \subseteq \Gamma$. $E_C = \int_{\bar{\Gamma}}\|\delta\bar{\mathbf{x}} - \delta\mathbf{x}\|^2\,\mathrm{d}s$ is the energy associated with the deviation from the prescribed displacement. The task in editing is to find a parameter update $\delta\Theta$ inducing the displacement $\delta\mathbf{x}$ which minimizes this energy. Since a parameter update can cause movement only in the normal direction, it can only approximate the normal component of the target.

$$\delta\Theta = \operatorname*{argmin}_{\delta\tilde{\Theta}} E_C = \operatorname*{argmin}_{\delta\tilde{\Theta}} \int_{\bar{\Gamma}}\|\delta\bar{x}_n - \delta x_n + \delta\bar{x}_t - \delta x_t\|^2\,\mathrm{d}s$$
$$= \operatorname*{argmin}_{\delta\tilde{\Theta}} \int_{\bar{\Gamma}}\|\delta\bar{x}_n - \mathbf{b}^\top\delta\tilde{\Theta}\|^2\,\mathrm{d}s + \int_{\bar{\Gamma}}\|\delta\bar{x}_t - \delta x_t\|^2\,\mathrm{d}s$$

(4)

In practice, the displacement is provided as a finite set of surface vectors at the locations $\{\mathbf{x}_i\}_{i=1..I} \subset \bar{\Gamma}$ and the integral is estimated with the sum:

$$\delta\Theta = \operatorname*{argmin}_{\delta\tilde{\Theta}} \sum_i \|\delta\bar{x}_n(\mathbf{x}_i) - \mathbf{b}(\mathbf{x}_i)^\top\delta\tilde{\Theta}\|^2 .$$

(5)

This is a linear-least-squares problem $\mathbf{B}\delta\Theta = \delta\bar{\mathbf{y}}$ with $\mathbf{B} = [\mathbf{b}^\top(\mathbf{x}_i)]^\top \in \mathbb{R}^{I \times P}$ and $\delta\bar{\mathbf{y}} = [\delta\bar{x}_n(\mathbf{x}_i)] \in \mathbb{R}^I$.

To improve the numerical stability one often uses Tikhonov regularization which penalizes the norm of the solution $\min_{\delta\Theta} E_C + \lambda\|\delta\Theta\|^2$ for some small positive regularization constant $\lambda$. In our setting, Tikhonov regularization serves an additional purpose: small $\delta\Theta$ are necessary for the validity of the linear expansion in Equation 1. Furthermore, especially in semantic editing, we might sample less points than parameters $I < P$, which would lead to an underdetermined system if regularization is not used. Lastly, regularization can also control how similar the final shape is to the source shape, as indicated in Figure 9.

**Target Deformation** We sample points $\mathbf{x}_i$ on the boundary $\bar{\Gamma}$ via iterative rejection sampling on the domain of interest or near the farthest point samples of the existing points. This is a sufficiently efficient method for our needs, although more advanced methods exist (Hart et al., 2002). Target deformations are then prescribed on the sampled points. If these are not given as the magnitude along the normal deformation $\delta\bar{x}_n$, but as a vector $\delta\bar{\mathbf{x}}$, we project them $\delta\bar{x}_n = \mathbf{n}^\top\delta\bar{\mathbf{x}}$. With partially prescribed targets, one must be careful about the target being unintentionally restrictive if the normals at the sampled points are nearly orthogonal to the target vector. This can be remedied by additionally filtering the points based on their normals.

**Large Displacements** Despite the boundary sensitivity in Equation 1 being valid only for small displacements due to the assumption of locally constant gradients, large displacements can be achieved via several iterations. The initially sampled surface points are moved by the computed $\delta\mathbf{x}$ in order to obtain the samples of the next iteration. To obtain the respective target deformations we split the initially user-provided target (either scalar along normal or vector) into equal parts. If the target is a vector, for each iteration we project the divided target vector onto the current normal. In the demonstrated geometric and semantic editing applications a small number of iterations (<15) is sufficient. Note, that the number of iterations will in general scale with the magnitude of the desired deformation.

**Constraints** Furthermore, we demonstrate the use of boundary sensitivity to fix a value of a functional during parameter updates. If the functional is expressed as a surface or volume integral,

this can be done without computing the integrals themselves. An example of this is constant area, which can simulate the behaviour of unstretchable, but bendable materials, such as rope in 2D or textile in 3D. Similarly, volume preservation is characteristic to incompressible materials (Desbrun & Gascuel, 1995) and is desirable in smoothing to prevent shrinkage (Taubin, 1995).

To this end, we loosely introduce shape derivatives and refer to Allaire et al. (2021) for more detail. $H'(\Omega)(\delta\mathbf{x})$ denotes a *shape derivative* of the functional $H(\Omega) \in \mathbb{R}$ if the expansion $H(\Omega_{\delta\mathbf{x}}) = H(\Omega) + H'(\Omega)(\delta\mathbf{x})$ holds for small $\delta\mathbf{x}$. We rewrite this using perturbations as $\delta H(\Omega)(\delta\mathbf{x}) = H(\Omega_{\delta\mathbf{x}}) - H(\Omega)$. For several functionals the shape derivatives are known analytically. For the functional $H(\Omega) = \int_\Omega h \, dx$ defined as a volume integral of $h = h(\mathbf{x}) \in \mathbb{R}$ the shape derivative is

$$\delta H(\Omega)(\delta\mathbf{x}) = \int_\Omega \mathrm{div}\,(h\delta\mathbf{x})\; \mathrm{d}x = \int_\Gamma h\delta\mathbf{x}^\top\mathbf{n}\, \mathrm{d}x \tag{6}$$

where the last equality is due to the divergence theorem on a bounded and Lipschitz domain $\Omega$. Inserting the boundary sensitivity from Equation 2 we arrive at

$$\delta H = \delta\mathbf{\Theta}^\top \int_\Gamma h\mathbf{b}\, \mathrm{d}x = \mathbf{b}_H^\top\delta\mathbf{\Theta} \tag{7}$$

where $\mathbf{b}_H := \int_\Gamma h(\mathbf{x})\mathbf{b}(\mathbf{x})\, \mathrm{d}x \in \mathbb{R}^P$ again acts as a basis, but for the perturbed integral quantity. $\delta H = 0$ can now be enforced either as soft constraint or as a hard constraint by projecting any parameter update $\delta\mathbf{\Theta}$ onto the $\mathbb{R}^{P-1}$ linear subspace where the integral stays constant $\delta\mathbf{\Theta}_{\delta H=0} = (\mathbf{I} - \mathbf{b}_H\mathbf{b}_H^\top)\delta\mathbf{\Theta}$.

Analogously, for a functional $G(\Omega) = \int_\Gamma g \, dx$ defined as a surface integral of $g = g(\mathbf{x}) \in \mathbb{R}$ the shape derivative as a perturbation is $\delta G(\Omega)(\delta\mathbf{x}) = \int_\Gamma (\partial g/\partial\mathbf{n} + \kappa g)\,\delta\mathbf{x}^\top\mathbf{n}\, \mathrm{d}x$, where $\partial g/\partial\mathbf{n}$ is the directional derivative of $g$ along $\mathbf{n}$ and $\kappa = \kappa(\mathbf{x}) = \mathrm{div}(\mathbf{n}(\mathbf{x})) \in \mathbb{R}$ is the mean-curvature. The corresponding basis for $\delta G = \mathbf{b}_G^\top\delta\mathbf{\Theta}$ is $\mathbf{b}_G = \int_\Gamma (\partial g/\partial\mathbf{n} + \kappa g)\,\mathbf{b}\, \mathrm{d}x$.

## 4 APPLICATIONS

Having established the framework for editing implicit shapes through boundary sensitivity, we consider its applications. We emphasize geometric and semantic editing since our framework unifies both while being agnostic to the architecture and training of the model. We also address deformation-rigidity-based editing since it classically is a widely used approach to shape editing. Lastly, we demonstrate the use of constraints with an example of volume-preserving smoothing.

In all cases, we include Tikhonov regularization with $\lambda = 0.1$ (see Appendix B). However, the behavior of regularization depends on the number of points $I$ and the magnitude of the prescribed displacement. In turn, $\lambda$ influences the number of required iterations and the residual.

### 4.1 GEOMETRIC EDITING

In *geometric* editing, the displacement is prescribed on the entirety of the boundary. In this section, we focus on studying the case where the prescribed displacement is local and the rest of the boundary is fixed, akin to local control in classic representations. Deformation in Section 4.3 and smoothing in Section 4.4 are examples of geometric editing with a global target displacement.

For each considered shape, we train a separate network to fit the value and surface normals of the SDF. All networks share the same architecture: 3 hidden layers of 32 neurons each with $\mathrm{sin}$ activations. In total, there are $P = 2273$ learnable parameters, all of which are manipulated during editing. We demonstrate several examples of geometric editing in Figure 1 where we displace parts of both man-made and organic neural implicit shapes. In addition, we quantify and plot the relative geometric error between the computed shape and prescribed target normalized by the largest target displacement $(\delta x_n - \delta\bar{x}_n)/\max_{\mathbf{x}\in\Gamma}|\delta\bar{x}_n|$. When the displacements are, loosely speaking, *natural* to the shape, the approximations recover the target well. However, not arbitrarily complex displacements can be approximated as in the example of inscribing letters. The characteristic length of the target is much smaller than that of the shape, especially in the relevant region. We hypothesize that the good memory-expressiveness trade-off of neural fields requires the NN to allocate geometric complexity

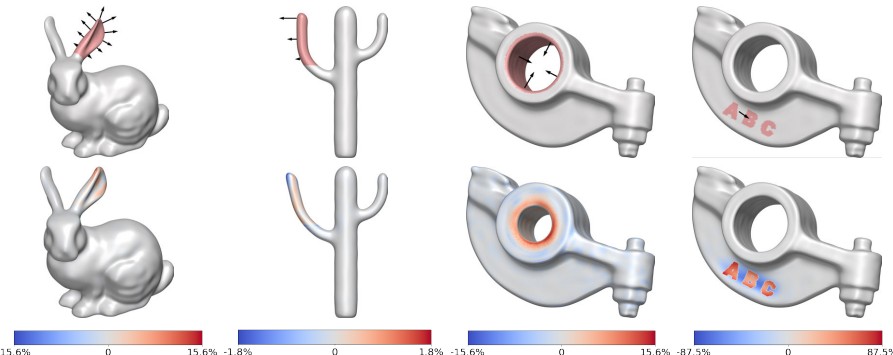

Figure 1: Geometric editing. Top: non-zero target displacement is prescribed on the highlighted regions and the remaining boundary is fixed with $\delta \bar{x}_n = 0$. Bottom: the resulting shape and normalized error. Natural displacements are approximated well. A counter-example is given in the last column: the prescribed displacements (imprinting letters) are too complex and outside the limits of expressible deformations, resulting in a coarse dent in the general area.

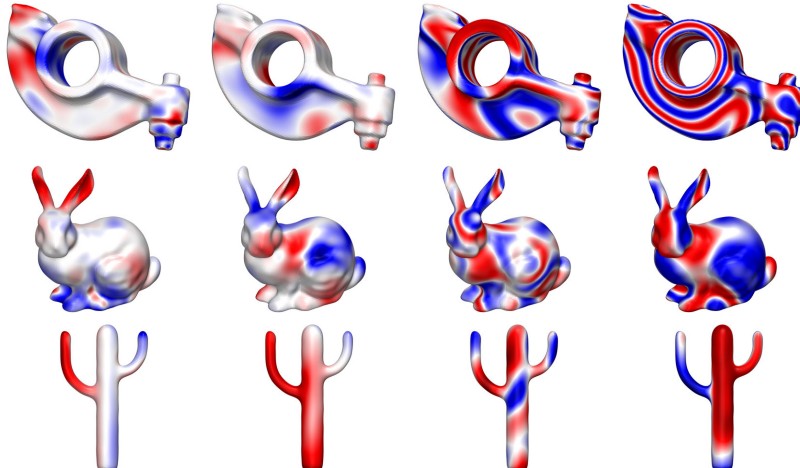

Figure 2: Basis functions of several neural implicit shapes trained on a single geometry. Each column depicts a weight from each layer increasing in depth left-to-right. Red/blue denote positive/negative deformation w.r.t. the outward normal. These bases do not strongly encode semantic or geometric meaning, different from latent basis functions in generative models in Figure 3.

where it is needed during training. To illustrate this, some basis functions are shown in Figure 2. As can be seen, their characteristic length is similar to that of the shape features. As all possible edits are a combination of these basis functions, it is unlikely that modifications on a much smaller scale can be reconstructed faithfully.

## 4.2 SEMANTIC EDITING

In representation learning the aim is to explain observations with a small number of *latent variables* (Bengio et al., 2013), which we will denote by l. Generative models $f(\mathbf{x}; \mathbf{l}, \mathbf{\Theta})$, such as (variational) auto-encoders or generative adversarial networks, attempt to map novel latent codes to novel outputs within the same distribution as the observations. When interpolating between two latent codes, the appearance of the generated output changes continuously (Shen et al., 2020a).

For a generative neural implicit shape model, the basis for the boundary movement is locally described by $\mathbf{b} = -\nabla_{\mathbf{l}} f / \|\nabla_{\mathbf{x}} f\|$ (Equation 3). Figure 3 illustrates a few of such basis functions for the generative decoder of the IM-Net model (Chen & Zhang, 2019). The decoder is trained as part of an auto-encoder reconstructing the entire ShapeNet dataset (Chang et al., 2015) from $\mathbf{l} \in \mathbb{R}^{256}$

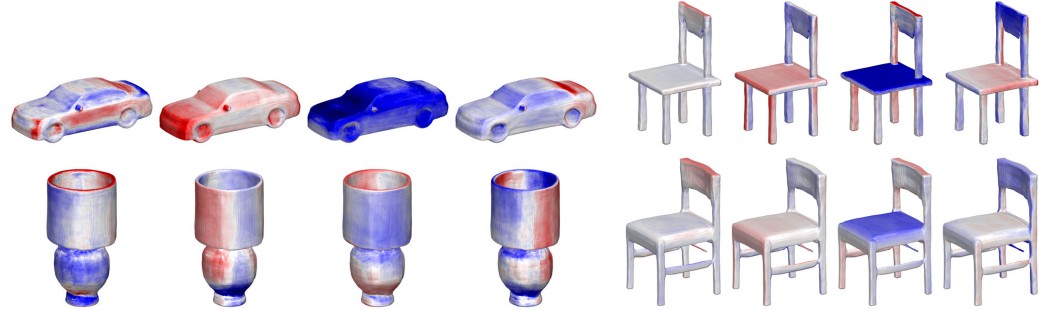

Figure 3: Basis functions of IM-Net. They partially encode semantics, such as segmentation and symmetries. Each column represents the same basis function. For similar shapes, such as the two chairs, the bases are qualitatively similar. Red/blue denote positive/negative deformation w.r.t. the normal.

latent variables. We use a pretrained model available at https://github.com/czq142857/IM-NET-pytorch, which is implemented as an MLP with 8 layers of 1024 neurons each and leaky-ReLU activations.

These basis functions can be observed to encode semantics, such as rotation and reflection symmetries, and segment semantically meaningful parts, such as windscreen on the car, rim of the lamp and different surfaces on chairs. For similar parts, such as the two chairs, the bases show similar behaviour.

However, it is known that not all directions in the latent space are (equally) viable (Chen et al., 2022; Vyas et al., 2021), hence, a similar argument can be made for the bases. Furthermore, while each basis function might cause meaningful deformation, they are not necessarily disentangled, i.e. each basis function does not explain a single generative factor, unless latent disentanglement is used in training (Bengio et al., 2013; Shen et al., 2020b; Tschannen et al., 2018). This could further increase the interpretability of the basis.

Despite this, with our method we can intuitively traverse the latent space by considering the link between geometric and latent variable changes. We present several examples in Figure 4, where the highlighted areas $\bar{\Gamma}$ of an initial shape are prescribed a local, high-level displacement, such as shortening a leg of a chair or squeezing the sides of the boat. We sample roughly 100 points in these areas, prescribe the same target vector at each point, and leave the remaining boundary unconstrained. After projecting the target onto the current normal and finding the best fit parameter update according to Equation 5, we repeat this process for a few ($< 15$) iterations to achieve visually obvious changes. User input is provided only at the beginning. Note that we only consider the latent parameters for our basis and leave all network parameters unchanged. Despite only prescribing local geometric deformation, we observe global and semantically consistent changes. Not only are obvious symmetries preserved, but also the morphology of the shape can change significantly, such as with the boat.

In Appendix A, we repeat the same set of experiments with the DualSDF (Hao et al., 2020) architecture. Each model is trained on a single ShapeNet category. This gives better generative results and a more semantically pronounced basis.

Compared to geometric editing, semantic editing has the computational advantage of sampling points on just a small part of the boundary $\bar{\Gamma}$. Furthermore, the number of latent parameters is much smaller than the number of NN parameters, leading to very small least-squares systems. Per iteration, the method only requires a single forward- and backward-pass through the model, altogether being fit for interactive use.

### 4.3 RIGID EDITING

As a final approach to editing, we briefly address deformation-rigidity since it classically is a widely used approach to shape editing. On the one hand, deformation energy is one of the many alternative priors to semantic prior discussed previously. A simple approach is to consider pure bending energy

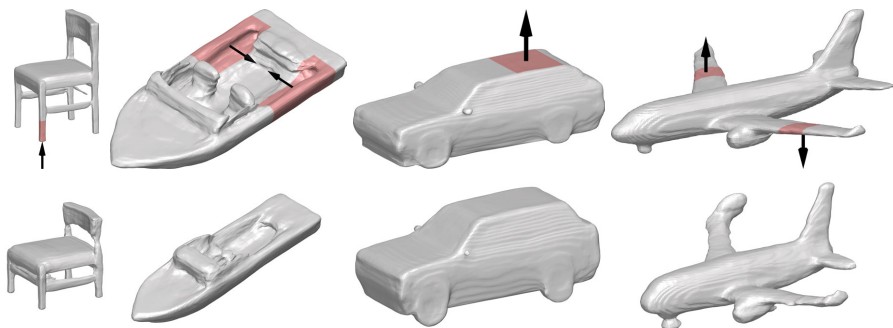

Figure 4: Semantic editing. Top: non-zero target displacement is prescribed on the highlighted regions $\bar{\Gamma}$ and the rest is *unconstrained*. Bottom: the result after $< 15$ iterations. A semantically plausible result is produced from a plausible locally prescribed displacement. A counter-example is given in the last column: prescribing each wing to move in opposite directions is not a semantically viable displacement and the method fails to approximate the displacements and produce meaningful results.

since it can be computed only from the implicit representation itself. On the other hand, typical applications, such as as-rigid-as-possible (Sorkine & Alexa, 2007) or as-Killing-as-possible (Solomon et al., 2011), also include the stretching energy. This is not trivially applicable to implicit surfaces, as there is no natural notion of stretch due to ambiguity in the tangent directions. As discussed in Section 3, there are several choices for recovering tangential displacements, but we consider perhaps the simplest one – using the tangential projection $\delta \mathbf{x}_t(\mathbf{x}; \Xi) = \left(\mathbf{I} - \mathbf{n}(\mathbf{x})\mathbf{n}(\mathbf{x})^\top\right) \mathbf{f}_t(\mathbf{x}; \Xi)$ of a separate sufficiently smooth NN $\mathbf{f}_t(\mathbf{x}; \Xi) : \mathbb{R}^3 \mapsto \mathbb{R}^3$ parameterized in $\Xi$.

To quantify the deformation energy, we leverage Killing vector fields (KVFs) which generate isometric deformations (Solomon et al., 2011). $\delta \mathbf{x}$ is a KVF if it has an anti-symmetric Jacobian $\mathbf{J}(\delta \mathbf{x}) = \mathbf{J}(\delta \mathbf{x})(\mathbf{x}) \in \mathbb{R}^{3 \times 3}$ everywhere on the boundary: $\mathbf{J}(\delta \mathbf{x}) + \mathbf{J}(\delta \mathbf{x})^\top = 0 \ \forall \mathbf{x} \in \Gamma$. A vector fields' deviation from being Killing can be measured with its *Killing energy* $E_K(\delta \mathbf{x}, \Gamma) = \int_\Gamma \|\mathbf{J}(\delta \mathbf{x}) + \mathbf{J}(\delta \mathbf{x})^\top\|^2 \, \mathrm{d}x$.

Classically, the Jacobian is computed using a discrete operator on a spatial discretization, while Atzmon et al. (2021) formulates AKVF on a neural implicit directly. We follow this approach and seek an AKVF respecting the prescribed boundary deformations $\delta \bar{\mathbf{x}}(\mathbf{x}_i)$ weighted by some small $\alpha > 0$: $\min_{\delta \tilde{\mathbf{x}}} E_K + \alpha E_C$.

Boundary sensitivity allows to search for the energy minimizing deformation directly in the space of expressible deformations and to directly optimize the normal component over $\delta \Theta$ using Equation 2. The $E_C$ in Equation 4 is expressed trivially but can now also accommodate the tangential component. To express $E_K$, we can leverage $\mathbf{J}$ being a linear operator $\mathbf{J}(\delta \mathbf{x}) = \mathbf{J}(\delta \mathbf{x}_n) + \mathbf{J}(\delta \mathbf{x}_t)$. Here $\mathbf{J}(\delta \mathbf{x}_n) = \mathbf{J}(\mathbf{b}^\top \delta \Theta) = \mathbf{J}(\mathbf{b}^\top)\delta \Theta$ with $\mathbf{J}(\mathbf{b}^\top) \in \mathbb{R}^{3 \times 3 \times P}$ and $\mathbf{J}(\delta \mathbf{x}_t) = \mathbf{J}(\mathbf{f}_t)$ straight-forward.

Figure 5 illustrates the results of AKVF deformation. Comparing these results with a mesh-based method, after sufficient iterations we achieve qualitatively similar results and comparable energies: $E_K^{\text{mesh}} = 254$, $E_K = 320$ for the bunny and $E_K^{\text{mesh}} = 11$, $E_K = 6$ for the cactus. However, due to the need to train an additional NN and perform second-order differentiation, our approach is about an order of magnitude slower than the mesh-based LSTSQ solver.

## 4.4 VOLUME PRESERVING SMOOTHING

As an example of fixing the value of a surface or volume integral, we fix the volume itself during smoothing. Fixing the volume of an implicit shape is difficult without resorting to an intermediate representation, such as a mesh (Remelli et al., 2020; Mehta et al., 2022), since there is no trivial way to differentiably compute the volume of an implicit shape. Smoothing on neural fields has been previously formulated using gradient descent on an objective penalizing the deviation from a specified mean-curvature (Yang et al., 2021) and as mean-curvature flow computed on an intermediate mesh (Mehta et al., 2022). We also use mean-curvature flow without resorting to the intermediate mesh, but

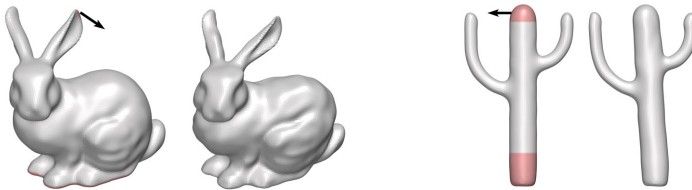

Figure 5: Rigid editing. Target displacement is prescribed in the highlighted regions $\bar{\Gamma}$ and the rest is unconstrained. In both cases, the bottom of the shapes is anchored.

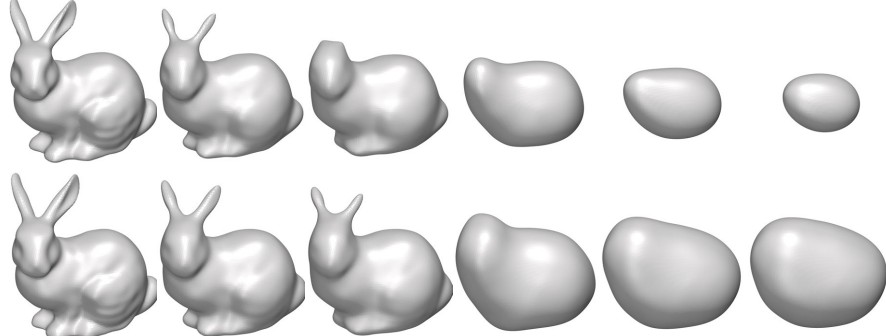

Figure 6: Select iterations of smoothing via mean-curvature flow without and with a volume constraint. The volume after 86 iterations decreases by $80\%$ and $1.6\%$, respectively. The constraint is achieved by projecting the parameter update onto the locally linear isochronic subspace found using boundary sensitivity.

the proposed volume-constraint is agnostic to the choice of the smoothing method. We first compute the mean curvature by expanding $\kappa(\mathbf{x}) = \mathrm{div}(\mathbf{n}(\mathbf{x}))$. From there, we can reuse the geometric editing framework with the globally prescribed target displacement $\delta x_n(\mathbf{x}) = -\kappa(\mathbf{x}) \, \forall \, \mathbf{x} \in \bar{\Gamma} = \Gamma$. Note, that this approach induces the correct geometric flow as discussed by Mehta et al. (2022). To fix the *volume V* while smoothing, we simply set $h = 1$ in the bases introduced in Equation 7. We enforce volume preservation by projecting the parameter update onto the isochronic subspace. Figure 6 compares smoothing with and without this constraint. After 86 iterations, the volume decreases by $1.6\%$ with and $80\%$ without the constraint. Ultimately, the shapes converge to a sphere and a singular point, respectively.

## 5 CONCLUSIONS

With implicit neural shapes becoming a widespread representation, we have demonstrated a unifying approach to perform geometric and semantic editing without the need for tailored training or architectures, while being simple to implement, and, especially in the case of semantic editing, fast and fit for interactive use. While we touched upon optimizing directly in the deformation space with a rigidity-prior, using other priors for unconstrained and tangential deformations remains an interesting problem. We used signed-distance and occupancy fields, but the editing framework can also be extended to other neural fields, where NeRFs especially provide tantalizing options. We hope that formulating a basis for the deformation space allows future work to further study and build models with desirable properties, such as interpretability, tailored degrees-of-freedom, linear-independence, and compactness or use them for segmentation or symmetry detection.

### ACKNOWLEDGMENTS

This was supported by the European Union's Horizon 2020 Research and Innovation Programme under Grant Agreement number 860843.

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

## A  SEMANTIC EDITING WITH DUALSDF

We repeat the experiments from Section 4.2 with a different architecture, namely, DualSDF (Hao et al., 2020). DualSDF learns a joint latent space for a fine-scale shape and its coarse approximation as a union of spheres. When a sphere is manipulated, an optimization process is run to find a latent code that explains the new sphere configuration, i.e. coarse shape. From the updated latent code the fine-scale shape can be generated.

Different to IM-Net, DualSDF is trained on individual ShapeNet categories. We use pretrained models on planes and chairs available at https://github.com/zekunhao1995/DualSDF. In addition, we train another model on cars following the provided training procedure on all 3515 car shapes available after preprocessing the ShapeNet category.

Figure 7 shows a few select basis functions of the three different DualSDF models. Figure 8 show the semantic editing results with our method, compared with the method described in DualSDF. For both methods, the shapes are generated with the same generative network. The difference lies only in how the edit is prescribed. In our approach we prescribe the movement directly on the sampled surface, while in DualSDF we move the spherical primitives, selecting them according to a similar design intent.

Both DualSDF and our method achieve plausible, though different, results. Both methods fail to adhere to the semantically implausible updates prescribed in the last column. However, the main benefit of our approach is that we are able to apply such manipulation to arbitrary NNs, whereas DualSDF needs a second NN for the coarse approximation and task-specific training.

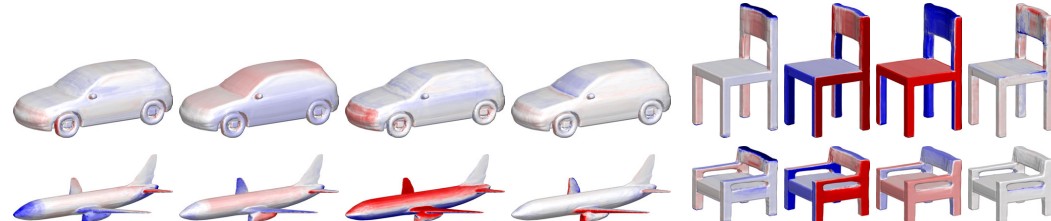

Figure 7: Select basis functions of three different DualSDF models. The semantics in most basis functions are not as prominent as the ones shown here. The two sets of chairs show the same respective basis functions, which can be seen to have the same semantic quality.

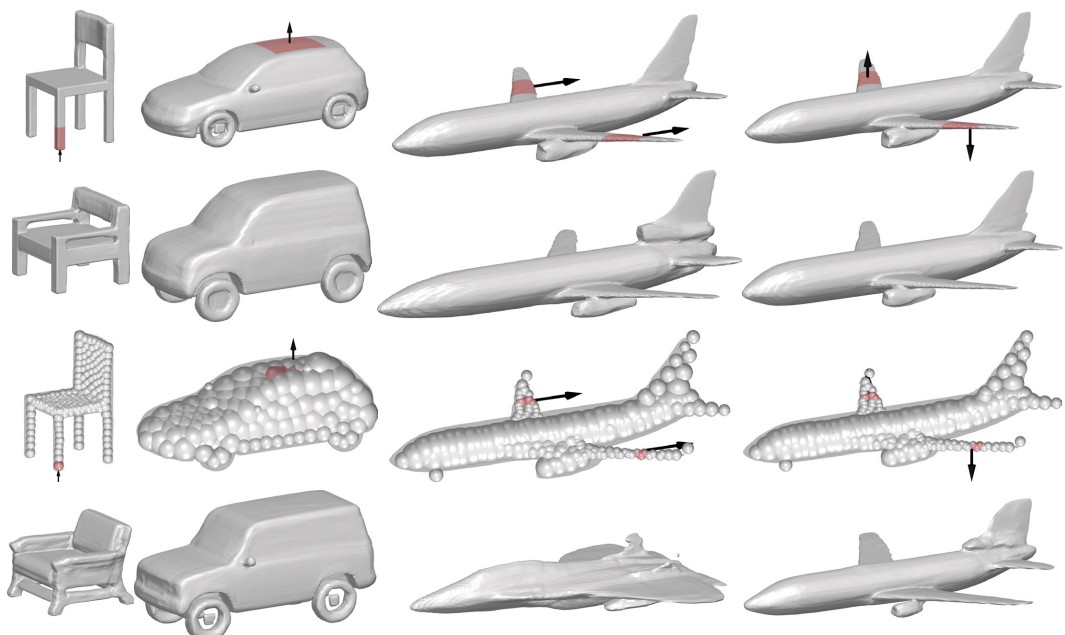

Figure 8: Semantic editing. The first two rows are similar to Figure 4, but use the generative network of the DualSDF architecture. The third and fourth rows show the prescribed deformation and the result using the editing procedure as described by Hao et al. (2020).

## B  EFFECT OF TIKHONOV REGULARIZATION ON SEMANTIC EDITING

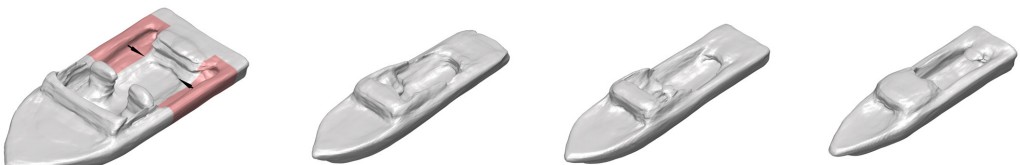

Figure 9: Effect of Tikhonov regularization on semantic editing. Left: source shape is prescribed an inward displacement on the highlighted regions while the rest is unconstrained. The three figures on the right have decreasing amount of Tikhonov regularization $\lambda = 10^1, 10^{-1}, 10^{-3}$. Stronger regularization better preserves similarity to the source shape, which is especially noticeable on the unconstrained front of the boat remaining wider. Stronger regularization also requires much more iterations to converge to a similar result (860 for $\lambda = 10^1$ compared to $< 10$ for the other two) since the the constraint violation $E_C$ is weighted less heavily than regularization.

## C  SPLITTING LARGE DEFORMATIONS

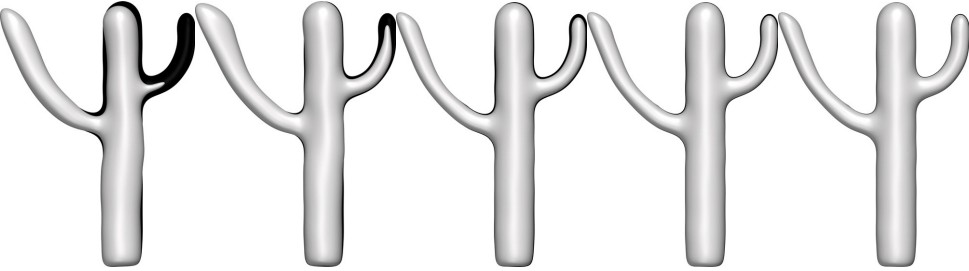

Figure 10: Effect of splitting a large target deformation in geometric editing. Displayed are the results after splitting the target into 1,2,4,8,16 equal parts. As the figure illustrates, this helps accurately recover large deformations which violate the first-order approximation. For comparison, the black silhouette in the background shows the target.

