# OpenReview forum: "Neural Implicit Shape Editing using Boundary Sensitivity"
_ICLR.cc/2023/Conference — ICLR 2023 poster_

### Official Review · Reviewer_mGb1 · 2022-10-23

**Confidence:** 4
**Correctness:** 4
**Technical Novelty And Significance:** 2
**Empirical Novelty And Significance:** 2
**Recommendation:** 6

**Clarity, Quality, Novelty And Reproducibility:**

Very clear paper, reproducible (apart for experiment 4.3 involving the tangential component as mentioned above). However, not much novelty.

**Strength And Weaknesses:**

The theory supporting the framework is very sound, clear, and easy to follow. As opposed to previous works deriving similar formulas ([1] Controlling Neural level sets, Atzmon et al., NeurIPS 2019 and [2] MeshSDF, Remelli et al., NeurIPS 2020), authors here explicitly write the the tangential component before zeroing it out.

As experiments demonstrate, it effectively works for shape modifications.
The provided visualization of “basis functions” (ie. ~gradients at intermediate layers) are nice and shed a light on what the network has learned, but cannot be used for any practical purpose.


My main concern is that this whole idea of finding gradients for network parameters corresponding to some shape modifications is not novel. [1] and [2] derived the exct same gradient, and [3] Sketch2Mesh, Guillard et al., ICCV 2020 already used it for interactive shape manipulation. The proposed method loses some generality compared to these frameworks which express gradients and chain rule for any downstream gradient, not only displacements.

The only 2 minor differences compared to [1] and [2] are:
- 1: [1] and [2] use gradient descent to update parameters, here a linear least square is used.
- 2: [1] and [2[ always zero out the tangential component of the gradient, here it can be non zero.

To convince me of the relevance of their approach over [1,2], authors would need to argue and demonstrate at least one advantage. It could be in terms of:
- Speed? Would linear least square faster than gradient descent?
- Using a non zero tangential component? This is only used in one experiment, not very well explained (I could not reproduce), and not ablated.
- Not having to reconstruct the explicit surface? But I think this is a misleading argument, because to pick a place where displacements must be applied, one must get an explicit representation of the surface first. And the reprojection of points sounds very much like an adaptive remeshing algorithm. Even more so if that was true, what would be the point of avoiding an explicit mesh in the loop? Speed? Marching Cubes is very fast with coarse to fine grids. Precision? Marching Cubes precision far exceeds the accuracy of the upstream MLP network…

Another minor concern is that it is not very clear to me how the displacement is indicated on the shape.


**Summary Of The Paper:**

This submission proposes a mathematical framework for editing 3D represented as deep implicit surfaces. Given a target displacement on the shape, the corresponding modifications of network parameters is computed and applied.
This can either be applied to a network representing a single shape (shape editing) or to the latent code of a multi-shape network. In the latter case, the network acts as a learned prior and allows semantic editing.


**Summary Of The Review:**

Nicely executed paper, but I don’t see the added value compared to ([1] Controlling Neural level sets, Atzmon et al., NeurIPS 2019 and [2] MeshSDF, Remelli et al., NeurIPS 2020) or even [3] Sketch2Mesh, Guillard et al., ICCV 2020

---

> ### Author Response · Authors · 2022-11-16
> **Response to Reviewer mGb1**
>
> ## Related work
> First of all, thank you for pointing out the two works that we had previously missed. We have incorporated Controlling Neural Level Sets [1] and Sketch2Mesh [3] in our related work section.
> We also updated it to more clearly reflect that we indeed are not the first to consider boundary sensitivity for different intents.
>
> Although [1] also derives boundary sensitivity, its use mainly focuses on classifiers and in the single considered geometric context on improved training. As such, we would argue that our focus on geometric applications is a notable difference.
> Sketch2Mesh [3] addresses a similar application as our semantic editing, but requires a dedicated data-set and model.
>
> ## Intermediate meshing
> To summarize, Sketch2Mesh [3], MeshSDF [2], and Mehta et al. [4] all use a mesh as an intermediate representation.
> Classic marching cubes meshes the whole surface, which is not required in semantic editing. While it would be possible to mesh just the relevant parts of the surface, we believe it is conceptually and computationally simpler to just sample points, especially since the prescribed handles can be single points.
> For the relatively large IM-Net model, we found that even hierarchic marching cubes was prohibitively slow or failed to accurately recover thin features. Admittedly, we did not try much more advanced mesh extraction techniques due to concerns on the ease-of-use and the sufficiency and simplicity of point sampling.
> Similar to some previous works in this field, we believe that a line of work exploring mesh-free approaches complements the intermediate mesh methods and is relevant especially in cases where differential geometric quantities (like curvatures) are required.
> This becomes more relevant with the development of real-time neural field rendering.
> Especially with this, but even by displaying just the sampled points or point-cloud based rendering, one would not require a mesh to visualize the shape and certainly not to prescribe the target deformations.
>
> ## Novelty
> With that, we would like to reiterate on the novelty in the following points.
> Formulating the terms in boundary sensitivity as basis offers intuition and a tool to study the expressible deformation space, thus relating neural to classic implicit representations. While we did not study this in sufficient detail, we hope this opens a future line of work with some ideas described in the conclusion of our work.
> Furthermore, we use boundary sensitivity to explicitly compute a basis for the perturbed integral quantities, like the volume, which in turn enables to project onto constraint sub-spaces, which to the best of our knowledge has not been demonstrated before.
> Similarly, a model- and training-agnostic semantic editing has not been shown before, and enables users to manipulate existing models without the need for any training.

---

### Official Review · Reviewer_TumX · 2022-10-25

**Confidence:** 4
**Correctness:** 4
**Technical Novelty And Significance:** 3
**Empirical Novelty And Significance:** 2
**Recommendation:** 5

**Clarity, Quality, Novelty And Reproducibility:**

While I really like the idea behind this work and I think it could have been an
amazing paper, I think the writing of the paper could be improved, since in its
current state many implementation details are not properly discussed.
While the authors do not introduce a novel methodology, the idea of using
boundary sensitivity to edit implicit shapes is novel, to the best of my
knowledge.

**Strength And Weaknesses:**

## Strengths:
-------------

1. I really like the fact that the proposed method is model-agnostic and can be
applied to a pre-trained Neural Network to deform a shape in place.

2. The idea of employing boundary sensitivity to associate changes in the
parameters of a previously trained model with the implicit shape is novel and
has many potential applications to various tasks.

3. The proposed idea is very simple and potentially can be applied for editing
various implicit-based shapes.


## Weaknesses:
-------------

1. While I really like the idea of this paper, unfortunately the experimental
evaluation is quite weak. No baselines are considered and no quantitative
metrics are reported, hence making it quite hard to evaluate of final edits.
While qualitatively the results are very impressive, I believe it is necessary
to also include some quantitative metrics that measure the realism of the final
shape. One potential baseline to consider, which is not 100% comparable with
the proposed method is the Deformation Handles. While Deformation Handles
cannnot operate on the implicit shape it might be interesting to see whether
they can yield similar results when applied to the reconstructed shape

2. Various details from the Applications section are missing. For example, in
section 4.1, the authors state "The considered models are trained to fit the
SDF and surface normals a shape and share the same architecture: sin activation
functions and 3 hidden layers of 32 neurons each." This sentence is a bit
unclear and contains some typos so I am not really sure what is the
experimental setup in this section. Do the authors train their model on each one of
the shapes in Figure 1? What is the input? How are the displacements
parametrized? Αre they defined only on the normal direction?

3. Another point that is a bit unclear from the text is whether the editing
operations are defined in relation to some target shape. I think that the
authors only define a target deformation rather than a target shape. Can the
authors please clarify this?

4. Another limitation of this work is that the authors do not experiment with
various network architectures. I think it would be useful to showcase that
their model can be applied on various implicit-based architectures such as
OccNet, DeepSDF, NeUS etc.


**Summary Of The Paper:**

The key idea behind this work is to leverage boundary sensitivity to express
how the perturbations in the parameters of a previously trained implicit-based
model alter the shape boundary. The authors consider two types of editing
operations: (i) geometric editing, which applies a geometric update on the
entire shape boundary and (ii) semantic editing, which applies a target
displacement only to a part of the boundary. The paper does not compare to any
baseline, nor reports any quantitative metrics to evaluate the quality of the
editing operations.

**Summary Of The Review:**

While I really like the idea of this paper, I am a bit concerned regarding the experimental evaluation, since the authors do not compare to other methods nor report any quantitative metrics. Moreover, since various details are unclear I am leaning towards rejecting the paper. Next, I have included various comments and questions regarding the paper:

1. In the 4th paragraph of Section 1.0, the authors state "the user supplies a
target displacement on (a part) of the shape boundary". This sentence is a bit
unclear. How is the target displacement defined/parametrized? While additional
details are provided later in the text, I think it is important to add one/two
more sentences at this point to elaborate more on this.

2. The Related Work section can be improved. For the first two groups of
papers, i.e. Implicit Shape Representations and Manipulation and Neural Fields,
I think it would be nice to clearly state that these works are not related work to the
proposed method. Instead the proposed method can edit implicit shapes generated
from these models by applying perturbations on the NN parameters. Regarding the
Neural Shape Manipulation section, I believe it would helpful to clearly
explain how boundary sensitivity is deployed in the works of Atzmon et al.,
Remelli et al. and Mehta et al., as stated in the last sentence of this
section. This analysis is important in order to better understand what is novel
in this work and what was already possible in prior research.

3. For the bottom right experiment in Figure 1, what is the editing operation
that is considered? Moreover, I don't fully understand why the ABC letters
disappear?

4. I think that the experiment in Figure 2, is not properly discussed. Can the
authors explain what was the message they intended to convey with this
experiment?

5. I think it would be great if the authors could also show results on the
geometric shape editing task on the IM-NET model that is used in the
experiments of Section 4.2. Being able to edit implicit shapes of a model
trained on multiple objects is a more challenging setup.


Typos:
-----
- Caption in Figure 2 has typo: depthg -> depth

---

> ### Author Response · Authors · 2022-11-16
> **Response to Reviewer TumX**
>
> We thank the reviewer for the suggestions on clarity and experimental evaluation. Hopefully, most of the suggestions are covered by the improvements described in the main answer.
>
> ## Comparison
> As mentioned in the general answer, we plan to add some further experiments.
> We will use DualSDF as a baseline comparison to our semantic editing.
> However, it seems to us that a quantitative evaluation here is not possible due to the semantic nature of the experiments, but we are open to suggestions.
> We will, however, add a quantitative evaluation of geometric deviation from the target for the geometric editing experiments.
>
> ## Diverse Network Architectures
> The mentioned networks are in our view quite similar as they all are MLPs. Thus, we feel it is redundant to evaluate them all (additionally, many of the works do not publish pre-trained weights and training such generative models from scratch is unfortunately quite wasteful).
> The additional experiment with DualSDF hopefully will be enough to showcase the use of a different MLP architecture and output (SDF), as opposed to IMNet with the characteristic function output.
> We think this covers the two flavors of MLP based architectures and would leave more exotic architectures (such as hybrid grid based or NERFs) for more application-specific future work, noting, that we do not see any reason on why these should not work.
>
> ## Geometric Editing in Latent Space
> Our interpretation of point 5 of the summary is that you propose to perform editing in the latent space of IM-Net, but with constraints over the entire surface of the shape.
> When it comes to editing in the latent space the quality of the results depends on how well the prescribed deformation agrees with the network's prior.
> This can be seen in the failure case in figure 4, where an "unnatural" deformation is prescribed and as a result editing fails to apply it.
> For geometric editing this will be more severe, as the deformations are more detailed and thus harder to account for with the limited degrees of freedom given, when we only modify a latent vector.
>
> ## Clarity
> We do indeed prescribe a target deformation, not a target shape in all of our experiments, although a target shape can be constructed from the target deformations (but not necessarily vice-versa).
> We hope that we were able to clarify this and all other questions in our revised version.
> We revised the structure of our paper (including the related work section) and added more explanations to unclear topics.

---

### Official Review · Reviewer_n2KN · 2022-10-26

**Confidence:** 4
**Correctness:** 3
**Technical Novelty And Significance:** 3
**Empirical Novelty And Significance:** Not applicable
**Recommendation:** 5

**Clarity, Quality, Novelty And Reproducibility:**

## Novelty
- The paper is based on classical sensitivity analysis for implicit surfaces, which is not new.
- The application to neural fields is also not new (Mehta et al. 2022), though its use without any mesh may be new.

## Clarity
- The manner in which user input is provided is unclear. In section 3, "Let $\delta \overline{\mathbf{x}}$ ... be a prescribed deformation on (a part of) the boundary" suggests that the user input is provided only at points on the surface. However, points on the surface are sampled with rejection sampling. So presumably the user input must be provided on a region of positive volume, in the neighborhood of part of the surface. This needs to be more carefully explained.
- The iterative scheme presented in section 4 needs to be explained better. Is new user input provided at each iteration? Or is the iteration meant to converge to the user's target displacement? How many iterations are needed?
- Section 4.4 is really about the method and should be moved to the end of section 3 and integrated with section 3. The second paragraph of section 4, describing the iterative displacement scheme, should also be moved into section 3 and expanded (see above).
- In the "semantic editing" application presented in Figure 4, it is not clear from the discussion whether the method optimizes the latent codes and weights together, or only the weights.
- In equation (8), it is unclear how $\int_\Omega \mathrm{div}(h\delta\mathbf{x}) \mathrm{d}x$ becomes $\delta \mathbf{\Theta}^\top \int_\Omega \mathbf{b} \mathrm{d}x$.
- The paper claims "the upper limit to how fine an arbitrary detail anywhere on the shape can be is given by $A/P$ where $A$ is the surface area of the shape" (p. 5). This certainly seems intuitively reasonable, but to claim it as certain more justification should be presented.

## Minor points
- In Figure 3: "show segmentation" is unclear
- Page 6: remove "Since this is valid for interpolation to many samples,"
- Page 6: "each basis" => "each basis function" (2x)
- Page 6: "allows to intuitively traverse" => "intuitively traverses"
- $\mathbb{R}^{3,3}$ => $\mathbb{R}^{3\times 3}$ and similarly for $\mathbb{R}^{3,3,P}$
- Page 7: "several choices to recovering" => "several choices for recovering"



**Strength And Weaknesses:**

## Strengths
- The paper is mostly clear and concise
## Weaknesses
- The paper has no quantitative empirical results. As such, it is unclear how well, e.g., volume preservation works. Since the method is only preserving volume to first order by using step-projections, this should be evaluated.
- Ditto for the AKVF approach to rigidity

**Summary Of The Paper:**

The paper presents a unified story for shape derivatives of neural implicit surfaces. The method enables projection onto deformations that preserve volume or are approximately rigid, and it suggests a path toward enforcing other kinds of constraints. Results are limited and only qualitative.

**Summary Of The Review:**

The paper's fully-implicit approach to editing neural fields seems valuable. More careful evaluation and explanation is required to make this a great paper.

---

> ### Author Response · Authors · 2022-11-16
> **Response to Reviewer n2KN**
>
> We thank the reviewer for the extensive feedback and have integrated most of it into our revision. We will try to clarify some further questions here.
>
> ## Quantitative Evaluation
> In fact we do give results for volume preservation in the text of section 4.4 (after the plotted 86 iteration, 1.6\% decrease in volume compared to 80\% and ultimately converging to singular point without constraints). To improve visibility, we also added this note to the caption of the figure.
> If a more thorough analysis is wished for, we can offer to repeat this experiment for several shapes and report results in the appendix.
> For the evaluation of AKVF we are somewhat unsure what a convincing quantitative evaluation would be.
> We can report the loss, but do not consider this insightful.
> Another option is to perform another AKFV deformation where the normal deformation is also represented with an independent NN (similar to tangent deformation) instead of the linear perturbation space. However, this would only quantify how restrictive the linear perturbation space is.
> Ideally, we would like to add a comparison to a mesh based AKVF until the revision deadline, or until the final deadline at the latest.
>
> ## Characteristic length of possible edits
> We do believe our claim that the level of detail is bounded by $A/P$ to be correct, as for smaller edits we would not have enough basis functions to edit the entire shape. However, we think this is not relevant enough to warrant a detailed explanation, therefore we removed the claim and added a more intuitive reasoning for the failure of small scale edits.
>
> ## Clarity
> We have addressed all mentioned points. Most changes are addressed in the general answer.
> Additionally, as you rightfully notice, equation (8), now (6), did indeed have an error in the integration domain and we have added a further note on the divergence theorem.
>
> ## Novelty
> Please refer to the answer to reviewer mGb1 for a detailed discussion on novelty.

---

### Official Review · Reviewer_VBEG · 2022-10-26

**Confidence:** 4
**Correctness:** 4
**Technical Novelty And Significance:** 3
**Empirical Novelty And Significance:** 3
**Recommendation:** 6

**Clarity, Quality, Novelty And Reproducibility:**

The paper is fairly clear. The work appears to be high-quality and original.

I was curious if large user-specified local edits can be handled by dividing them into small incremental changes and then time-stepping the method, i.e. to move the bunny's ear by 10 units, break it into 10 steps of 1 unit each and recursively apply the method to obtain the final integrated deformation of the bunny.


**Strength And Weaknesses:**

The paper is well-written and develops an interesting method for computing the boundary sensitivity of an implicit field. Various applications and examples are presented to highlight the utility of the approach. In general, I am positive about this paper.

One weakness is that the paper does not really go into how and where the method fails. E.g. for what magnitude of displacements does the linearity assumption lead to incorrect results? If the network is very complex (very high-dimensional), are the geometric edits induced by perturbing network weights still smooth enough to look plausible? How does this latter aspect play with the explicit constraints induced in Sections 4.3 and 4.4?

Another weakness is that it does not sufficiently compare to baselines, as pointed out in other reviews.

**Summary Of The Paper:**

This paper introduces a method for interactively editing 3D shapes defined as the level sets of neural implicit functions. Specifically, the approach quantifies the distribution of changes to this level set (the shape boundary) w.r.t. the parameters governing the neural implicits (either the network weights for a model regressed to fit a single shape, or the latent variables of a generative model). This is referred to as "boundary sensitivity". Based on this, the authors demonstrate several example of shape editing guided by high-level sparse user input, as well as introducing deformation constraints (rigidity, volume preservation...) into the framework. While this sort of inverse control has been demonstrated for other representations, this is, to the best of my knowledge, a novel contribution for neural implicits.


**Summary Of The Review:**

I think the paper develops a nice mathematically justified approach for tying implicit field boundary changes to parameter updates, using this to solve various inverse control tasks. I think it can be better positioned w.r.t. evaluations and prior work.

---

> ### Author Response · Authors · 2022-11-16
> **Response to Reviewer VBEG**
>
> Thank you for the helpful feedback to our approach. We have already addressed some points in the main answer and will try to answer all remaining questions satisfactorily.
>
> ## Network Complexity
> For the presented constraints in general there is a $P-C$-dimensional sub-space available around an already feasible point that satisfies the $C$ constraints given $P$ degrees-of-freedom.
> Consequentially, the model complexity should not in principle impact the constraints themselves, but rather the objective that is optimized along the constraints.
> For such general objectives, such as in editing, a more complex network is in fact beneficial, as there are more degrees-of-freedom available.
> Hence, in geometric editing we would expect to approximate the densely prescribed deformation better with more complex networks.
> Similar argument holds for the approximation power of the sparse target in semantic editing with respect to the size of the latent code.
> Since the network parameters (weights and biases) remain fixed in the considered semantic editing experiments, the effect of the model complexity on editing would be indirect via the learned basis which depends on the training process.
> The problem lies in the other direction. If the model does not provide enough degrees-of-freedom, prescribed deformations are difficult to approximate.
>
> ## Displacement Iterations
> Splitting large deformations into several small ones is in fact how we deal with this problem. We made this clearer in the relevant section, and will add an explicit experiment on this in the appendix.

---

### Author Response · Authors · 2022-11-16
**General answer**

We thank the reviewers for their valuable feedback, which we have incorporated to the best of our ability into our revised document. We will address the changes we made in a little more detail. All relevant changes in the document have been color coded for easier review. Rearranged previous text is shown in orange and new additional clarification in blue.

## Changes
Based on the reviewers suggestions we have reworked some parts of our paper to increase the clarity.
Beside some unmarked more minor improvements, the major changes we applied are the following:
We have moved the introduction of the iterative editing algorithm into section 2 and give more information to its details.
- All formulas regarding shape derivatives are now split off from section 4.4 and moved to section 2. Section 4.4 thus now only contains the application part.
- For clarification we have added paragraphs explaining how the target deformations are given and how large displacements are handled.
- The related work section now contains a separate paragraph discussing all previous works that include boundary sensitivity.
- We extended some explanations regarding network training, characteristic lengths and semantic updates.

## Evaluation
For a stronger quantitative evaluation and comparison of our results, we are currently in the process of providing further experiments:
- We will quantify the error achieved in geometric editing and plot the errors in Figure 1.
- In addition to IM-Net, we will perform semantic editing on DualSDF. This addresses the suggestions of trying an additional architecture and having a baseline to compare with.
- We will add an experiment to show at what update magnitude the linearity assumption breaks down and the updates are no longer sensible.

We are confident, that we can add these results to the revision until the deadline, but wanted to submit a preliminary revision and comments to give you some time to respond to this already late rebuttal.

---

### Author Response · Authors · 2022-11-18
**Revision 2 for additional evaluation**

We have added a second revision addressing the suggestions for additional evaluation. In particular:
 - Figure 1 in Section 4.1 now quantifies the error for the geometric editing examples.
 - Appendix A adds an additional semantic editing experiment with a different architecture, as well as a baseline comparison to DualSDF.
 - Appendix C adds an experiment illustrating the effect of splitting large updates into several steps.

We have put the new experiments in Appendix for now as to preserve the existing structure for your convenience.
Due to time constraints we did not yet implement a mesh-based AKVF baseline, but plan to include this in the later version.

---

### Author Response · Authors · 2022-12-13
**AKVF baseline**

We have also implemented a baseline for the AKVF deformation on meshes extending to 3D the method described in _Solomon et al, As-killing-as-possible vector fields for planar deformation, Computer Graphics Forum, 2011_. We use a discrete gradient operator to compute the Jacobians and assemble and solve a sparse LSTSQ system. We use the original meshes (that were used for fitting the neural fields) instead of the ones extracted with marching-cubes, since these lead to very poorly conditioned systems which did not converge. Both the method discussed in the paper and the mesh-based method share the same deformation handles and use the mesh vertices as evaluation points to achieve better comparability.
Our method achieves qualitatively similar result after sufficient iterations. We use the mesh Jacobian to quantify the Killing energy for deformations computed with the two methods. The mesh-based deformation has lower energies: $E_K = 254$ versus $E_K=644$ after 5000 iterations for the bunny test case and $E_K = 11$ versus $E_K=20$ after 1500 iterations for the cactus test case. The Killing energy is somewhat smaller when computed with automatic-differentiation instead: $E_K=320$ and $E_K=6$ for the bunny and cactus, respectively. This indicates that the AD and mesh gradient operators differ slightly. In addition, we note that due to the need to train an additional NN and do second order differentiation, our approach is about an order of magnitude slower than the mesh-based LSTSQ solver.

---

### Decision · Program_Chairs · 2023-01-20

**Decision:**

Accept: poster

**Justification For Why Not Higher Score:**

Given that I (and reviewers) am conflicted about whether the paper should be accepted at all, I have a hard time justifying a higher score than a poster accept.

**Justification For Why Not Lower Score:**

The paper offers enough value to the research community that I can't say with confidence that it should be rejected.

**Metareview: Summary, Strengths And Weaknesses:**

This paper proposes a method for interactively editing shapes represented as level sets of neural implicit functions.

Strengths:
- Interesting method for computing boundary sensitivity of an implicit field
- Supported by sound theory
- Various applications presented to demonstrate usefulness of the approach
- Is (in principle) model-agnostic; can be applied to different neural field architectures

Weaknesses:
- No discussion of weak points / failure cases
- Doesn't really compare to any plausible baselines
- Model-agnosticism is not evaluated very much (only a couple of architectures are tested, each on a different application, so there's no side-by-side comparison).
- Authors missed a couple of very relative pieces of prior work that do something similar. To my eye, this is the most serious criticism raised by any reviewer, but this reviewer is actually currently slightly in favor of accepting the paper.

Reviewers did engage in discussion with authors during the discussion period. Still, there was no clear signal on whether reviewers were generally in favor of accepting or rejecting the paper, so I called for a meeting.

**Note From Pc:**

if the above contains the word "oral" or "spotlight" please see: "oral" presentation means -> notable-top-5% and "spotlight" means -> notable-top-25%. As stated in our emails, we are disassociating presentation type from AC recommendations

**Summary Of Ac-Reviewer Meeting:**

3/4 reviewers attended the meeting.

Reviewers thought that paper has really nice theory: the mathematical foundations are formalized better and explained more clearly here than in any prior work. That said, this idea is not as novel as authors originally thought: there are the two works that one of the other reviewers brought up that need to be considered. After extensive discussion, reviewers felt like (in addition to the nice theory), the paper has a couple of potentially novel aspects: being totally mesh free and the ability to (approximately) respect constraints (such as volume preservation). Reviewers were not very satisfied with the experimental support of these contributions, though: there weren't really any applications presented that would require the ability to avoid generating an intermediate mesh, and the constraint satisfaction (being approximate) was not sufficiently evaluated (e.g. it would've been good to produce a bunch of synthetic deformations of ShapeNet shapes and measure how little volume change resulted).

After the meeting, weighing all the evidence, opinion seemed to be trending toward reject as "the right thing to do," but one reviewer expressed some sadness at this prospect, believing that the nice mathematical formulations and results in this paper had value to the community even if the experimental evaluation could be better.

I sent the above text to the one reviewer who was not able to attend the meeting; they later sent me an email with the following comments:

"Regarding the paper, I am honestly quite troubled, hence I kept my initial rating to borderline. First of all, I think that already, after the revisions, the paper is looking much better and it definitely can be useful in future research as a model-agnostic tool to perform editing in the latent shapes. One of the main reasons, why I suggested to include more models in their evaluation was to showcase how general their method is. For example, does their model only work when the latent representation is a signed distance field, what happens if the network is not MLP-based? Unfortunately, this point was not addressed by the authors in the rebuttal... Regardless of this, I think that while the paper is already looking much better, the experimental evaluation is still a bit weak, as there are no quantitative comparisons/results. I understand that quantifying the quality of edits is not trivial. However, as this is their main task, I think it would have been beneficial to propose a metric to quantify this. But as I myself don't have a good idea/suggestion on what this metric should like I don't think it is fair, for the authors, to propose rejecting their paper solely based on this."

I am still somewhat conflicted about what to do with this paper. If forced to make a decision, I would say that on balance it is probably better to accept the paper: it would be nice to see more thorough evaluation, but it's not as if we would be introducing something potentially "wrong" into the academic literature by accepting the paper without this evaluation.